# VS-Cambium-Developer: A New Predictive Model of Cambium Functioning under the Influence of Environmental Factors

**DOI:** 10.3390/plants12203594

**Published:** 2023-10-17

**Authors:** Daria A. Belousova, Vladimir V. Shishov, Alberto Arzac, Margarita I. Popkova, Elena A. Babushkina, Jian-Guo Huang, Bao Yang, Eugene A. Vaganov

**Affiliations:** 1Research Department, Siberian Federal University, 660041 Krasnoyarsk, Russia; popkova.marg@gmail.com; 2Institute of Fundamental Biology and Biotechnology, Siberian Federal University, 660041 Krasnoyarsk, Russia; vlad.shishov@gmail.com; 3Institute of Ecology and Geography, Siberian Federal University, 660041 Krasnoyarsk, Russia; aarzac@gmail.com (A.A.); evaganov@sfu-kras.ru (E.A.V.); 4Khakass Technical Institute, Siberian Federal University, 655017 Abakan, Russia; babushkina70@mail.ru; 5College of Life Sciences, Zhejiang University, Hangzhou 310058, China; huangjg@scbg.ac.cn; 6School of Geography and Ocean Science, Nanjing University, Nanjing 210023, China; yangbao@lzb.ac.cn

**Keywords:** cambial activity, cell differentiation, Vaganov–Shashkin model, climatic factors, *Larix sibirica*

## Abstract

Climate changes influence seasonal tree-ring formation. The result is a specific cell structure dependent on internal processes and external environmental factors. One way to investigate and analyze these relationships is to apply diverse simulation models of tree-ring growth. Here, we have proposed a new version of the VS-Cambium-Developer model (VS-CD model), which simulates the cambial activity process in conifers. The VS-CD model does not require the manual year-to-year calibration of parameters over a long-term cell production reconstruction or forecast. Instead, it estimates cell production and simulates the dynamics of radial cell development within the growing seasons. Thus, a new software based on R programming technology, able to efficiently adapt to the VS model online platform, has been developed. The model was tested on indirect observations of the cambium functioning in *Larix sibirica* trees from southern Siberia, namely on the measured annual cell production from 1963 to 2011. The VS-CD model proves to simulate cell production accurately. The results highlighted the efficiency of the presented model and contributed to filling the gap in the simulations of cambial activity, which is critical to predicting the potential impacts of changing environmental conditions on tree growth.

## 1. Introduction

Annual rings on tree stems, roots and branches result from the cambial activity during the vegetation period (or growing season) in different climate conditions [1,2]. The cambium is a thin layer of small, thin-walled cells between the xylem and phloem, capable of multiple divisions [3]. Thus, the cells produced within a growing season by the division of the cambium form a tree ring [3,4,5]. Seasonal tree-ring formation involves interrelated processes at different levels: cell division, enlargement and differentiation, controlled by hormonal balance, whereas nutrient availability, cell growth and cell wall formation depend on internal biochemical processes [3,6,7], directly influenced by external climatic factors [8,9,10].

In conifers, the result of these processes leads to a specific cell structure within the ring, from wide and thin-walled earlywood cells to narrow and thick-walled latewood cells, oriented toward different functions (Figure 1) and influenced by the environment during their formation over the growing season [11,12]. Individual tracheids constitute up to 95% of the xylem in conifers [13], derived from the corresponding cambial zone and consisting of a single initial and several mother cells [14]. The production of mother cells in the xylem significantly exceeds that of the phloem (up to ten times) [2]. Therefore, it can be assumed that the number of xylem mother cells is significantly higher than the number of phloem cells.

The cell cycle is the basis for understanding the cambium functioning over the growing season [3]. Its main characteristic is the cycle duration, equal to the time interval between two consecutive divisions and its phases. In the cell-division cycle, a cell undergoes four phases (Figure 2):(1)Presynthetic phase (G1), where the cell increases in size and duplicates its cellular contents;(2)Synthesis phase (S), which is for DNA coping and synthesis;(3)Premitotic phase (G2), where the cell increases in size while organelles and proteins prepare for cell division;(4)Mitosic phase (M), where the cell partitions the two copies of the genetic material of the mother cell into the two daughter cells.

To the best of our knowledge, no single theory currently considers the regulation-level hierarchy or provides a basis for predicting the process variability under the influence of internal and external factors of cambial activity [3,7]. A broad range of the literature suggests that the balance of auxins and cytokinins may be involved in the regulation of cambial activity [15,16,17,18,19,20,21] as well as the balance of substrates (sugars) and enzymes [19,22,23,24,25]. Similar gradients have been observed for several cyclins [26,27,28,29,30]. Moreover, it also shows a relationship between the influence of external climatic factors and cell cycle gene expression [25,31,32,33,34,35].

Although several process-based tree-ring models for conifers have been proposed, in which growth is limited by water balance [36,37],organic compound balance [38,39], water and temperature regimes [40,41,42], stem damage modeling (taking into account growth limitation by hormonal balance) [43,44], the Turgor-driven growth model [45], the XyDyS model [46,47], as well as models of the cambium functioning in deciduous species [48,49,50,51,52,53], attempts to quantify cambial activity dynamics remain limited despite its importance [3,37,47,54]. Furthermore, most existing models are complex in structure and have high requirements for the quantity and quality of input parameters and variables. In most cases, such modeling requires a manual year-to-year calibration of the parameters, making it challenging to apply them as predictive models, alongside having severe limitations. For example, the model developed by Drew and co-authors [41] allows us to estimate the cambial activity only for “ideal” external conditions when thermal and water regimes are constant.

The existing cambial block of the Vaganov–Shashkin model (VS model) does not consider the influence of internal factors on the growth of the cambial zone [3,55]. The cambial block of this model and its new modification, the VS-band model of cambium development [56], do not require the manual calibration of parameters, year-to-year, for long-term cell production reconstruction and forecast of conifer species. The VS-Cambium Developer (VS-CD) is a deep modification of the cambial block of the VS model [55] and can be considered a transfer from modeling cell division under external climate, forcing to model using the principal mechanisms of cambial activity regulation and the production of new xylem cells [57].

This work is a complex upgrade of the VS-CD, where the growth inhibitor concentration in each cambial cell is dynamically changing. Assumptions of a downward gradient of a hypothetical substance (growth inhibitor) may also have analogues, e.g., the concentration gradient of the differentiation inhibitor [58], the gradient in auxin and cytokinin activity, the gradient in cyclin activity, etc. Thus, the new version of the model predicts the cell production of conifers year-to-year based on the influence of external climatic factors and simulates the dynamics of radial development in the growing seasons.

The study aims to create an adequate and experimentally verifiable model of cambium functioning without a manual year-to-year adjustment of parameters. In this paper, we argue that the cambial growth of *Larix sibirica* Ledeb. in cold semiarid conditions is mainly regulated by the influence of external climatic limiting factors. The new version of the model was calibrated and verified using anatomical data from the forest-steppe zone of southern Siberia. The new structure of the cambial block of the VS model improves the understanding of cambium functioning under the influence of external environmental factors and can potentially be used for long-term reconstructions of cell production, a key component in tree-ring formation.

## 2. Results

The algorithm simulated the development of the cambial zone in each growing season with parameterization in automatic mode. The following steps were performed:An “average growth season” was simulated for the selected tree sample, and average tree-ring indices and cell production for all available growth seasons were calculated.The VS-CD model parameters for the “average growth season” were adjusted to obtain optimal simulation results for the average production and time of initial cell division.Parameters constant from season to season were applied to simulate cell production of the selected tree in each season.The inhibitor stock for each season was manually calculated from 1963 to 2011, in automatic mode, using a linear regression model.

The average cell production (22.7 cells) was calculated by direct anatomical data over the 1963–2011 period. Based on the VS model environmental block outputs, the average start and end of the growing season were in the DOY (day of the year) 127 and 285, respectively (Appendix A). The following seasons were removed from further analysis:(1)When cell production data were not prepared (1978 and 2003);(2)When the significant growth discrepancy between simulated and actual tree-ring indices was observed to exclude incorrect input data in VS-CD model calculations (1977, 1988, 2001, 2008; see Appendix A).

VS-CD model parameters were estimated over the obtained average growing season based on the average integral growth rate over 1963–2011 (see Appendix A).

The parameters showed in Table 1 were constant or invariant to seasons.

Inhibitor supply for individual seasons was estimated manually and in automatic mode using the linear regression (Equation (1)). There were two criteria: cell production modeling accuracy and season length modeling adequacy.

The algorithm for the calculation of cell production in automatic mode was verified in two steps:Verification of the algorithm for the equivalence of the automatically calculated inhibitor supply to the manually calculated inhibitor supply, per season.Verification of the algorithm by cell production, estimated with automatically calculated inhibitor supply.

For seasonal inhibitor supply calculations, the calibration period was defined as 1989–2011 ($R^{2}$ = 0.58), and the verification period was 1963–1987 ($R^{2}$ = 0.41). For cell production simulations, the calibration period was 1989–2011, ($R^{2}$ = 0.54), and the verification period was 1963–1989 ($R^{2}$ = 0.49) (Figure 3).

A significant positive correlation was seen between the simulated cell production based on and the observed values ${(R}^{2}=0.51,\;R=0.715,\;p<0.01)$(Figure 4).

According to the VS-CD model, the growing season is defined by the period of complete consumption of the inhibitor supply by the cambial cells. Compared to the VS-CD model, the VS model environmental block overestimates the length of the growing season by 35.5 days, on average.

Figure 5 shows a typical cambial kinetics of one season (1971) simulated by the model (Figure 5a). Correlations were calculated for spline-smoothed kinetics. The correlation between the dynamics of the cambial cell growth rate in phase G1 and the dynamics of the external growth rate, calculated by the VS model environmental block, was 0.77 ($R^{2}$ = 0.592). The correlation between the dynamics of inhibitor in cells and the dynamics of external growth rate, calculated by the VS model environmental block, was 0.58 ($R^{2}$ = 0.597). The correlation between the dynamics of inhibitor in cells and the dynamics of cambial cell growth rate in G1 phase was 0.74 ($R^{2}\;$= 0.818) (Figure 5b).

## 3. Discussion

The VS-CD model accurately simulates the cell production in *Larix sibirica* trees over a 49-year period. The results highlight the efficiency of the developed model, contributing to filling the gap in the simulations of cambial activity, which is critical for predicting the potential impacts of changing environmental conditions on tree growth in the future.

Previous versions of the VS model have been broadly used to simulate tree-ring growth for conifer [3,10,12] and broadleaf species [59]. Several attempts have also been devoted to simulating of cambial activity [3,55], including direct observations [60]. A previous version of the VS-CD model was applied to simulate the growth of cambial zone cells under the influence of external climatic factors with high accuracy [57]. However, it was required to manually adjust three (out of thirteen) parameters to calculate the growth rate of cambial zone cells within a season. In the new version of the model, the growth rate for each cell is calculated in automatic mode and only depends on the concentration of inhibitor in the cell and the external influence of climatic factors, reducing the time necessary for simulations. The automatic annual adjustment was performed for the one parameter specifying the inhibitor supply in the initial cell at the start of the growing season.

Moreover, the start of cambial activity simulated by the VS-CD matches the start of the growing season, estimated by the widely used VS model [10,12,60], determined by two criteria: (i) reaching the threshold temperature sum for the period and (ii) reaching the minimum threshold temperature for growth [10,55,60]. The end of the growing season (DOY) is considered when the total supply of the inhibitor is consumed, or the end of the growing season day calculated by the VS model environmental block is reached. Our results suggest that the VS-CD model underestimates the end of the growing period compared to the calculations of the VS model environmental block, i.e., the inhibitor’s supply in a season was consumed completely. On average, the difference was 35.5 days. It could be explained by the fact that the VS-CD model does not consider the processes of cell enlarging, cell-wall thickening, and cell lignification over the growing season. It has been shown that the VS model environmental block may overestimate the end of the growing season up to 19 days for the conifers in Canada [12], or remote sensing observations in southern Siberian Scots pines [61]. Therefore, the VS-CD can be considered a tool to refine cambium phenology in the future.

The growth rate of cambial cells in the G1 phase and inhibitor consumption during the seasons were both simulated correctly, which was verified by comparing the dynamics of daily growth rates provided by the VS model environmental block outputs. There was a highly significant correlation between both dynamics rates (R = 0.769, *p* < 0.00001). In addition, periods of climate stress cause a decrease in cambial cell growth rate.

Mother cell growth rate has a maximum at the start of the growing season and decreases toward the end of the season synchronously with the daily growth rates calculated by the VS model environmental block based on the climate data. This corresponds to the natural functioning of the cambial zone: in the absence of environmental stress, the cambial activity and radial growth rate of the xylem usually reach their maximum during the summer solstice, with a maximum photoperiod [3,7]. The inhibitor in cambial cells is consumed gradually during the season, and its concentration gradually decreases by the end of the season. There was a strong correlation between inhibitor consumption and external growth factors (R = 0.584, *p* < 0.01). Our results show that external conditions explain about 50% of the variation in cell production (Figure 4) in the cold semi-arid conditions of central Siberia.

The developed model is gradually moving from a process-based model to one that partially considers the main mechanisms regulating cambium activity and the production of new xylem cells. For a further improvement of the VS-CD model functioning and its verification, it should be contrasted with new direct and indirect observations of xylogenesis from different species and habitats. In the near future, the integration of VS-CD and VS-band models [56] is planned to clarify the processes regulating cambium activity and production of new xylem cells, as well as the integration of VS-CD model on R Shiny technology into vs-genn.ru web service. Finally, updated versions will be upgraded via widgets.

## 4. Materials and Methods

### 4.1. The VS-CD Model: Algorithm for Predicting Cell Production

Based on the cambial block of the VS model and the VS-CD, a new predictive algorithm of cambial functioning was developed to assess cambial activity without additional year-to-year parameter adjustment [57]. This model is based on the hypothesis that external climatic factors regulate the concentration of the differentiation inhibitor in each cambial cell, gradually decreasing with the distance from the initial cell. A differentiation inhibitor is defined as a hypothetical substance that promotes cambium formation. Practically, this can be expressed as the balance of auxins and cytokinins or the balance of substrates (sugars) and enzymes, etc. [19,62,63,64].

In the VS-CD model, the growth season length and the number of cells transferred to the enlargement zone (cell production) depend on the supply of the differentiation inhibitor in the initial cell at the start of the growing season. Therefore, in calculating the inhibitor supply, the model also considers the inhibitor consumption for tree adaptation to stress periods caused by growth-limiting climatic factors.

The start of the growing season and the periods of climate limitation during the season are estimated by the VS-CD model based on the daily integral growth rates estimated by the environmental block of the VS model [10,60,65,66,67]. The integral growth rate was determined by the minimum partial growth rates calculated for each limiting factor as a function of air temperature and soil moisture content weighted by the photoperiod [10,65]. In addition, the period of influence of external climatic factors was calculated for the cell cycle of each mother cell in the radial series and was considered if the daily integral growth rate consistently decreased for three days or more before the cell division [57].

The model inputs are described in Table 2. There are a total of 15 parameters, six of which are based on direct xylogenesis observations (cell sizes of mother and initial cells in every cell cycle phase, critical inhibitor concentration), and the user sets the approximate values manually by using actual anatomical measurements. The user defines only four parameters: (1) cell growth rate $V_{0}$ in the S, G2 and M phases; (2) the theoretical parameter of the growth rate in the G1 phase; (3) the coefficient of inhibitor distribution between daughter cells and (4) the stress factor parameter. The next four parameters—daily integral growth rates, length of the growing season, start and end of the growing season—are determined by the VS model environmental block outputs (for more details see [55]). The fifteenth parameter is the inhibitor supply, which is calculated automatically.

The inhibitor supply in the initial cell at the start of the growing season ${Inht}_{i}$ quantifies the tree’s ability to produce new cells under the influence of climatic factors, considering the consumption of useful substances to resist the stress factors in the previous season. The parameter ${Inht}_{i}\;$was calculated by the environmental growth rate curve and its statistical characteristics of the current and previous growing season due to the next equation:(1)$${Inht}_{i}=1.184+9.44*sd\left({Gr}_{i}\right)+2.0209*median\left({Gr}_{i}\right)-3.4693*mad\left({Gr}_{i}\right)-\;\;\;\;\;\;\;\;\;\;\;\;\;\;\;\;\;\;\;\;\;\;16.5541*\min\left({Gr}_{i}\right)-1.4214*max\left({Gr}_{i}\right)-0.3408*skew({Gr}_{i-1}),$$
where ${Gr}_{i}$ and ${Gr}_{i-1}$—daily growth rates in the growing season of current (i) and previous (i−1) seasons, respectively; sd—standard deviation; median—median; mad—median absolute deviation; min—minimum; max—maximum; skew—coefficient of asymmetry of previous season distribution. Regression coefficients were obtained using multiple linear least-squares regression in a model test on larch samples collected in southern Siberia [68].

The simulation starts with the first division of the initial cell for each season. Multiple divisions of the initial cell are possible during the growing season [3,55]. Initial and mother cells divide according to the same rules, producing two cells of equal size, half the size of the divided cell. A mother cell in the row loses its ability to divide and transfers into the enlargement zone, if the concentration of the inhibitor reaches the minimum threshold. This threshold is defined as an input model parameter. The initial cell ranks at position zero in the cell row, and the mother cells at positions from one onward. The inhibitor is distributed unequally among the daughter cells during each division. The inhibitor proportions are user-defined and they are fixed for all divisions within a growing season. The amount of inhibitor received by the daughter cells from the mother or initial cell is calculated as:(2)$$Inht\left(j\right)=\left\{\begin{array}{c} Inht\left(i\right)*\alpha \\ Inht\left(i\right)*(1-\alpha )\; \end{array}\right.,$$
where j is the position of the daughter cell in the radial file relatively to the initial, i is the position of the mother cell in the radial file relatively to the initial, $Inht\left(j\right)$ is the amount of inhibitor in the daughter cell in position j (relative value), $Inht\left(i\right)$ is the amount of inhibitor in the mother cell in position i (relative value) and α is the proportion of inhibitor. After division, the mother cell in the row disappears, and the daughter cells occupy positions j=i and j=i+1. The daughter cell receives a larger amount of inhibitor at the position closer to the initial cell (j=i). Compared to the previous version of the VS-CD model, the new one does not consider the stress factor when calculating the amount of inhibitor received by the daughter cell. See a visual output example of a cell dividing from VS-CD further below (Figure 6).

At the first step (i.e., the start of the growing season), there is only one initial cell in the cell row after the dormancy stage. The user sets its initial size in microns manually. The VS-CD model calculates the timing and rates of all cell cycle stages for the initial cell. The transition criterion for a cell from one phase to another is when the cell reaches a specific size. The cell growth rate in phases S to M is set manually by the user as model input, and the growth rate in phase G1 is calculated using the Gompertz function [69,70]:(3)$$v=\left\{\begin{array}{c} \frac{\left(D_{M}-D_{init}\right)}{d},\;to\;divide\;the\;initial\;cell\;\\ {\;e}^{{-e}^{-c*C_{inht}}},\;division\;under\;optimal\;conditions,\;\\ e^{{-e}^{-c*S*C_{inht}}},\;stress-induced\;division\; \end{array}\right.$$
where $D_{M}$ is the critical cell size for completion of the mitosis phase, $D_{init}$ is the initial size of the cell, d is the period of initial division, *c* is the cell growth rate parameter, *S* is the stress factor and $C_{inht}$ is a concentration of inhibitor in the mother cell in the phase of mitosis.

To calculate the concentration of inhibitor in the cell, the following equation was used:(4)$$C_{inht}\left(j\right)=\frac{Inht(j)}{D_{j}},$$
where *j* is the cell number in the radial row, $C_{inht}(j)$ is the inhibitor concentration (relative value) and $D_{j}$ is radial cell size (µm), Inht(j) is the amount of inhibitor received by *j*-cell from the mother cell.

The algorithm is completed if one of the conditions is reached:The length of the simulated growing season reaches the length of the season calculated by the VS model environmental block outputs;The whole supply of the inhibitor is consumed. In this case, the VS-CD model determines the season length.

The model outputs include the complete information on all divisions per season (see Table 3). The model records the dynamics in cell production and the changes in the duration of simulated growing seasons (see Figure 7).

The principal characteristics of the cambial activity within the growing seasons are the variation in the growth rate of mother cells in the G1 phase and the variation in the amount of inhibitor in mother cells. A season is correctly simulated if the cell growth rate corresponds to the dynamics of daily growth rates provided by the environmental block of the VS model. The initial growth rate is lower than the mother cell growth rate. Mother cell growth rate has a maximum at the start of the growing season and decreases toward the end of the season. The concentration of the inhibitor in cells gradually decreases by the end of the season [3,7].

The algorithm was implemented with the R language and environment for statistical computing and graphics [71,72] and uploaded at http://www.vs-genn.ru/VSCD2 (accessed on 1 October 2023; for all questions concerning the application launching and utilization, please contact the developer: Daria Belousova, daryadarya1611@gmail.com). The algorithm is a script whose output is only two types of data tables: for a whole time period and for a particular year (season) (Figure 8).

### 4.2. Comparison the VS-CD Simulations with Observed Anatomical Data

The VS-CD simulations were verified by the actual anatomical measurements of Siberian larch (*Larix sibirica*) trees growing in a monospecific forest stand in southern Siberia (54°00 N, 91°01 E, 660 m asl), located near the Sayan-Altay mountains and the Batenev Range in the forest-steppe zone, a region where the growing season begins in late April and ends in early October [73].

Wood cores were extracted from sixteen trees in 2012 at 1.3 m using a 5 mm borer (Haglof, Sweden) perpendicular to the direction of the slope to avoid reaction wood. Samples were processed according to standard methods of dendrochronology [7,74]. Tree-wring width was measured using the LINTAB system and crossdated in COFECHA [75,76].

Individual tree-ring chronologies were merged into the local chronology using the ARSTAN program [77,78]. A negative exponential curve was used for standardization. An autoregressive model was used to remove autocorrelations from the time series, which were averaged using a weighted robust mean estimate. A chronology reflecting the average response of trees across the site to the effects of external climatic factors was obtained.

Five trees, characterized by the highest correlation coefficients with the master chronology (0.6–0.8) and age over 100 years, were selected for anatomical measurements. Wood cores were softened by boiling in water prior to sectioning. Thick transverse sections (20 μm) were obtained using a sliding microtome (Reichert, Germany), and stained with safranin (1%). Transverse sections were digitized at 400× magnification under a microscope interfaced with a digital camera (AXIOCam MRc5, Axio Imager D1; Carl Zeiss, Germany). Cell number, radial cell size and cell wall thickness were measured in five radial files for each annual ring over the period 1963–2011 (see Appendix A). Measurements were performed using the image analysis package Lineyka, SuperMoment and ProcessorKR [79].

We used site-averaged chronology to model the average response of trees to the influence of climatic factors. To model tree growth indices using a VS-oscilloscope [67] (see parameters in Appendix A), climate data were used from the nearest weather station Minusinsk (53°41′ N, 91°40′ E, 254 m asl), mean annual temperature 1.0 °C and total annual precipitation 330 mm (http://meteo.ru, accessed on 1 October 2023). Smoothed temperature data obtained by moving the average method with a window of 11 days were used.

## 5. Conclusions

The results of this study showed that the VS-CD model was consistent with direct observations of the cell production in *L. sibirica* trees over 49 years in southern Siberia, and that cambium kinetics are linked to changes in climatic conditions.

This confirmed the efficiency of this version of VS-CD as a tool for data analysis in dendrochronology and filling the gap in the modeling of cambial activity kinetics using the VS model. The study contributes to a better understanding of the cell growth inhibitor’s nature based on carbohydrate balance and hormonal control under the influence of changing environmental conditions on tree growth.

## Figures and Tables

**Figure 1 plants-12-03594-f001:**
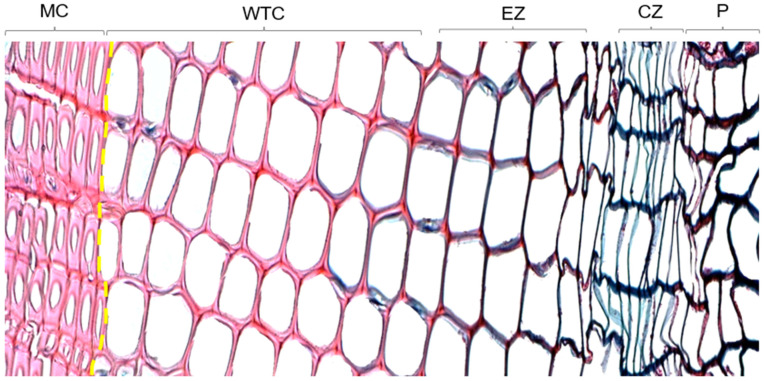
*Larix sibirica* thin section from southern Siberia showing different cell phases. Matured cells (MC), cell wall thickening phase (WTC), cells in the enlarging zone (EZ), cells in the cambial zone (CZ), phloem cells (P). Earlywood/latewood border is shown in yellow.

**Figure 2 plants-12-03594-f002:**
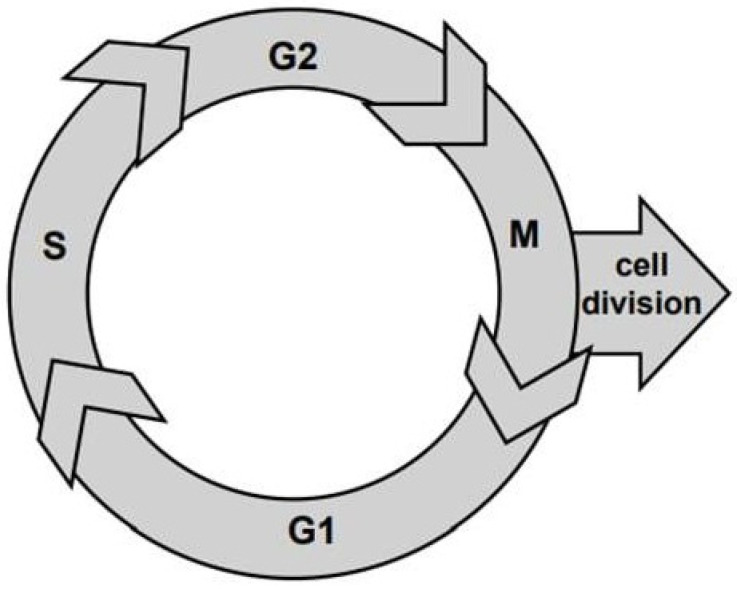
The cell-division cycle and corresponded phases: presyntetic G1, synthetic S, premitotic G2, and mitotic M.

**Figure 3 plants-12-03594-f003:**
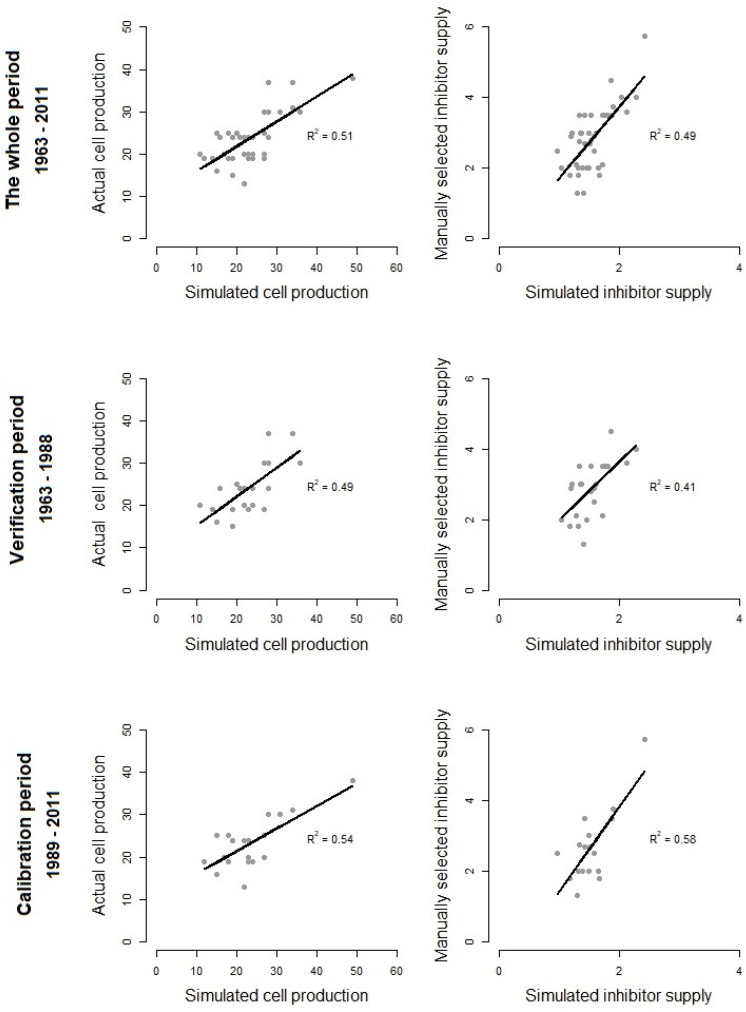
Calibration and verification of cell production and inhibitor supply calculations using the VS-CD model. Dependence of the observed and VS-CD-modeled cell production for 1963–2011 is displayed, as well as the results of calibration (1989–2011) and verification (1963–1989) of the results on these data. For the cell production simulation, the inhibitor supply was calculated in automatic mode using Formula 1.

**Figure 4 plants-12-03594-f004:**
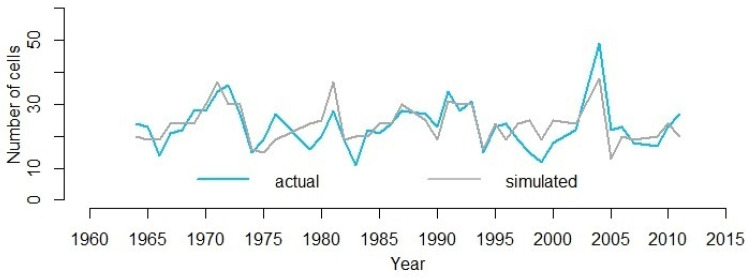
Simulation result of cell production, the gray curve shows the actual observations of cell production; the blue curve shows the result of simulation by VS-CD model.

**Figure 5 plants-12-03594-f005:**
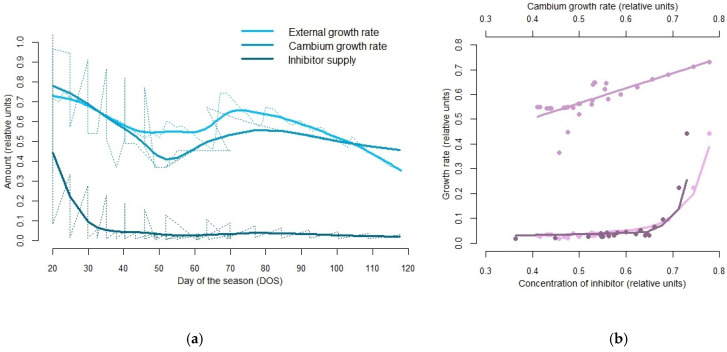
(**a**) The cambial kinetics for the simulated season of 1971. The external daily growth rates calculated by the VS model environmental block (light blue line), the cambium growth rates in the G1 phase (dark blue line), and the concentration of inhibitor in cambial cells (petrol blue line) are plotted as raw data and their approximation with a smoothing spline (bold curves corresponding colours). The first day of the season is considered the day when cambial zone cell growth starts. (**b**) Cross-correlations were calculated for the smoothed data (gradient purple scatterplots). The relation between the dynamics of the cambial cell growth rate in phase G1 and the dynamics of the external growth rate, calculated by the VS model environmental block, is linear (purple line in the top). The relation between the dynamics of inhibitor in cells and the dynamics of external growth rate, calculated by the VS model environmental block, is exponential (dark purple line in the bottom). The relation between the dynamics of inhibitor in cells and the dynamics of cambial cell growth rate in G1 phase is exponential (light purple line in the bottom).

**Figure 6 plants-12-03594-f006:**
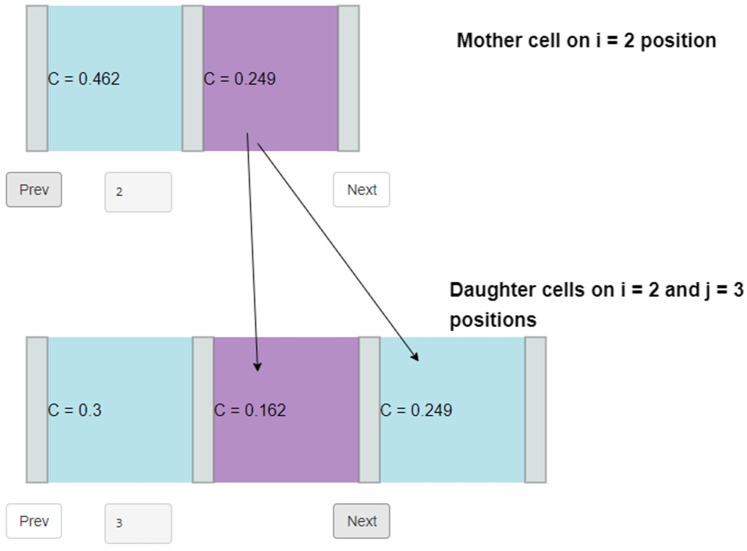
Visual output from VS-CD demonstrates the cell division from two cells in a radial file to three. Cell on *i* = 2 position divides into two cells on i = 2 and j = 3 positions. C describes an inhibitor concentration of a current cell in the current day [57].

**Figure 7 plants-12-03594-f007:**
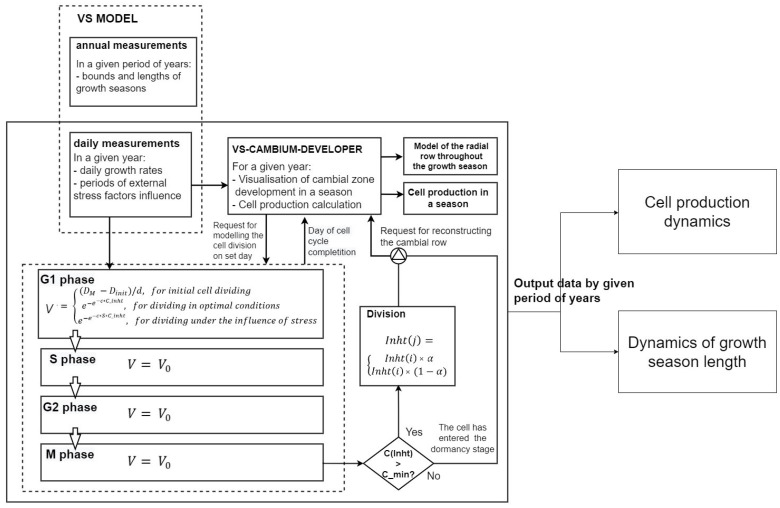
Visualization of the VS-CD model algorithm. Annual measurements include modeled growth indexes. Daily measurements include cell growth rates for each day in a season, calculated assuming the influence of external climatic factors only. The cell production from the VS-CD model is compared with actual observations, and the duration of the growing season is compared with an assessment of the duration of the growing season by the VS model environmental block outputs. The model functioning does not require annual manual calibration of parameters due to is performed automatically.

**Figure 8 plants-12-03594-f008:**
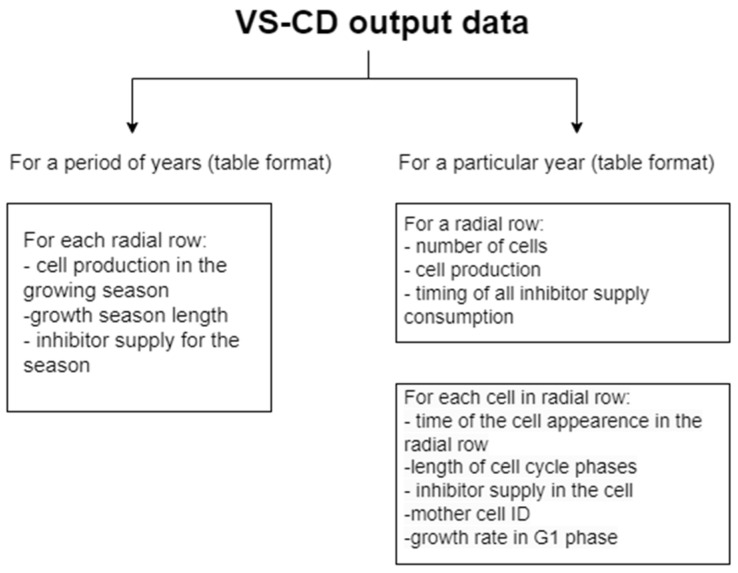
Visualization of the VS-CD model output data. The main module of VS-CD model executes R script and automatically creates and shows a summary table for a period of years and, at the user’s request, it automatically creates and shows a summary table for a particular year or for every year in a period, which includes information about all cells in the radial row.

**Table 1 plants-12-03594-t001:** Recorded invariant parameters of the VS-CD model.

Parameters	Dimension	Value
Critical cell diameter transitioning to phase S from phase G1	μm	8.5
Critical cell diameter transitioning to phase G2 from phase S	μm	8.7
Critical cell diameter transitioning to phase M from phase G2	μm	9.0
Critical cell diameter for division	μm	9.5
Minimum inhibitor concentration required for cell division	relative units	0.03
The growth rate V_0_ in S, G2, M phases	μm/day	0.68
Parameter of growth rate in phase G1	relative units	0.071
Diameter of initial cell	μm	2.0
Stress coefficient of growth rate in phase G1	relative units	100
Coefficient of inhibitor distribution between daughter cells	relative units	0.8286

**Table 2 plants-12-03594-t002:** Description of the VS-CD model input data and parameters.

N	Type	Parameters	Dimension	Description
1	Parameters and data calculated by the environmental block of the VS model (see [55,67] for more details)	Length of the growing season	day	Estimated by VS model and depends only on external climatic factors
2		Daily integral growth rates	relative units	Computed by VS model from daily data of temperature, precipitation and hours of sunlight
3		Start of the growing season	day	Determined when a specified sum of temperature is reached for a period
4		End of the growing season	day	Determined when the integral growth rate reaches a critical value
5	Fixed VS-CD model parameters for all seasons	Critical cell diameter transitioning to phase S from phase G1	μm	The positive approximate value is manually set using actual anatomical measurements
6		Critical cell diameter transitioning to phase G2 from phase S	μm	
7		Critical cell diameter transitioning to phase M from phase G2	μm	
8		Critical cell diameter for division	μm	
9		Minimum inhibitor concentration required for cell division	relative units	The positive approximate value is manually set
10		The growth rate V_0_ in S, G2, M phases	μm/day	The approximate value is manually set from 0 to 1 using VS model growth rates
11		Parameter of growth rate in phase G1	relative units	The approximate value is manually set (see “c” in Equation (3) below)
12		Initial cell diameter	μm	The approximate value is manually set using actual anatomical measurements
13		Stress coefficient of growth rate in phase G1	relative units	The approximate value is manually set (see “S” in Equation (3) below)
14		Coefficient of inhibitor distribution between daughter cells	relative units	The approximate value is manually set from 0.01 to 0.99
15	VS-CD model parameters representing the cell growth rate	Supply of the inhibitor in the initial cell at the start of the growing season	relative units	Calculated automatically (see Equation (1) below)

**Table 3 plants-12-03594-t003:** Description of output dataset for each division of the VS-CD model.

Parameters	Dimension
Date of current cell appearance in the season	Days (accuracy 0.001)
Timing of current cell phase G1	Days (accuracy 0.001)
Timing of current cell phase S	Days (accuracy 0.001)
Timing of current cell phase G2	Days (accuracy 0.001)
Timing of current cell phase M	Days (accuracy 0.001)
Timing of current cell cycle	Days (accuracy 0.001)
Amount of inhibitor received from mother cell	relative units
Current cell growth rate in phase G1	μm/day
Inhibitor concentration in M phase (needed to assess whether further cell division is possible)	relative units
Unique cell ID	text string
Mother cell ID	text string

## Data Availability

Data available on request from the authors.

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
