# Peer review of "VS-Cambium-Developer: A New Predictive Model of Cambium Functioning under the Influence of Environmental Factors"

_plants, 2023, doi:10.3390/plants12203594_

Round 1
Reviewer 1 Report
The investigation appears to be carefully designed and implemented. The paper is well written.
This reviewer has limited experience in inhibitor dynamics. Therefore, misunderstandings are possible.
The following remarks will hopefully be useful for the Authors in making the paper more understandable for readers with limited experience in the field.
The main problem for this reviewer is the following:
Are the ring indices predicted as a function of measured ring indices, using the Inhibitor supply from Eq. (1) as a detour? If so, what is the idea of predicting the values of a quantity by measurements of the quantity itself?
Line 16: are all internal factors of biochemical character?
Line 130: is it justify to remove observations not consistent with the applied model?
Line 293: this reviewer does not understand Eq. (2). Are the two lines somehow alternative? Should the parameter k appear in the Equation?
Reviewer 2 Report
The submitted work for review " VS-Cambium-Developer: A New Predictive Model Of Cambium Functioning Under The Influence Of Environmental Factors ", concerns the development of an algorithm for the prediction of wood growth under Cold Climates.
The topic of the research carried out is topical and important especially under conditions of a changing climate. Due to the increase in temperature in the northern hemisphere, it is to be expected that changing environmental conditions will have a significant impact on wood growth, which requires monitoring.
Simulation models are an excellent tool for assessing environmental changes, e.g. the description of wood increment in years, but can also be used for estimations in future conditions based on available climate scenarios.
I congratulate the authors on their idea. I very much like the concept of the paper, especially the idea of applying the physiological regularities of cell division to the estimation of cambium increment. That is, the development of the VS-CD-model for predicting cell production. Furthermore, I like the sequence of the tasks carried out in the work. From the presentation of the idea concept to the developed assumptions (algorithms) calibration, validation and verification of the results.
The literature presented demonstrates a broad knowledge of the subject matter.
A valuable aspect of the work is the chapter "Materials and methods" where the stages of the work performed are described in detail.
The results are statistically elaborated and presented clearly in graphs and described in detail. The only thing that puzzles me is why the study did not use the whole series of available data from the development of the algorithm since 1945 (Figure S2.).
In my opinion, the work deserves to be published.
Reviewer 3 Report
I consider the work to be interesting and appropriate for the journal, which is very technical, and the manuscript complies with that.
I have only one comment, it would be that they explain more the 15 parameters they used. They say they have six that measure directly but they don't mention exactly how. Adding all the parameters, I add 14 and not 15, so I would ask that they be explained a little more and there be a review of them. This comment is based on the content of lines 259-266.
Seems fine to me.
Reviewer 4 Report
Minor Comments:
1. Figure. 1. As the authors describes the earlywood/latewood in the main text (Line 42-43), so I suggest the authors mark the region of earlywood and latewood in this figure.
2. Line 295. the "Inht(k)" does not appear in function 2, is it wrong?
